# Relational Capital and Post-Traumatic Growth: The Role of Work Meaning

**DOI:** 10.3390/ijerph18147362

**Published:** 2021-07-09

**Authors:** Ting Nie, Mi Tian, Hengrui Liang

**Affiliations:** 1School of Business, Macau University of Science and Technology, Macau 999078, China; tnie@must.edu.mo; 2Independent Researcher, Xining 810016, China; henryliangt@gmail.com

**Keywords:** relational capital, post-traumatic growth, psychological security, work meaning

## Abstract

Through a statistical survey of 760 front-line medical staff during the COVID-19 epidemic, this study attempts to explore the relationships between relational capital, psychological security, post-traumatic growth and the meaning of work. Data analysis verifies that trust, reciprocity, and identification can promote post-traumatic growth by enhancing the individual’s psychological security. A high level of work meaning can enhance the role of trust, reciprocity and identification in promoting psychological security. Work meaning has a moderated mediating effect when trust and reciprocity affect post-traumatic growth through psychological security, but no moderated mediating effect is found when identification affects post-traumatic growth through psychological security.

## 1. Introduction

The epidemic in 2020 has caused the entire human world to face a challenge that has not been met for a century. As of July 2021, 205 countries around the world have been affected, with more than 186 million confirmed cases and more than 4.02 million deaths (Refer online real time data from https://www.outbreak.my/zh/world accessed on 8 July 2021). People all over the world are working hard to prevent and control the epidemic. With the spread of multiple shocks of COVID-19, the world economy is in recession, industrial development and business operations have been hit hard, and everyone’s lives have been affected. This has been and will be a challenge for the entire human race as well as for each individual.

In the fight against the epidemic, front-line medical staff have made tremendous contributions. However, as witnesses of the epidemic, they have been harmed both physically and mentally in the high-pressure environment of this disaster. The COVID-19 pandemic has led to a large increase in the incidence and prevalence of mental health problems, such as anxiety and depression [1], and is placing enormous stress on healthcare workers, whose mental health may have been damaged, caused by these events [2]. If medical staff are in a state of emotional exhaustion, they will naturally enter the resource preservation mode, and they are more likely to have work withdrawal behaviors the next day [3]. In particular, when the meaning of work is low, the COVID-19 crisis perceived by the health care workers increases, and their work engagement and responsibility decrease; therefore, the organization should carry out training and effective intervention to avoid deterioration of the situation [4]. The trauma brings not only harm, but many people also have re-considered their values, interpersonal relationships, and self-understanding. Because of the epidemic, some positive changes have taken place. In a sense, some have grown after trauma, due to disaster or adversity. Post-traumatic growth is exactly what contemporary positive psychology pays attention to today. Negative effects and positive effects after trauma coexist; various positive effects can promote the development of individuals and enhance their perception of happiness [5]. Previous studies have confirmed that when individuals have enough support, they can stimulate post-traumatic growth [6], but there are still some differences in the influence mechanisms of different types of support. Under the Chinese cultural background, we pay more attention to interpersonal relationships. The formation of trust, reciprocity, and identification within the organization create a special kind of capital, that is, relational capital [7]. Since COVID-related stressors can frustrate employees’ sense of belonging, increase their sense of insecurity, and have a negative impact on their behaviors [8], it is meaningful for us to explore whether mutual trust and cooperation within the organization and a high degree of identification can help medical staff have a positive sense of belonging so as to better engage in work and obtain growth opportunities during the COVID-19 crisis; namely, can relationship capital promote the occurrence of post-traumatic growth by improving the individual’s psychological security? In particular, when the individual has a high sense of work meaning, will this promotion effect be strengthened? This study selected medical staff that were on the front line during the outbreak of COVID-19 as the sample to explore the mediating role of psychological security between relational capital and post-traumatic growth, and to verify whether work meaning has a moderating effect.

## 2. Theoretical Basis and Research Hypothesis

### 2.1. The Impact of Relational Capital on Employee’s Psychological Security

Regarding relational capital, academics generally recognize that individuals establish relational asset through mutual communication, interaction, and cooperation, including the three dimensions of trust, reciprocity and identification [7]. The enhancement of relational capital can promote mutual cooperation within the organization and gain the support of other stakeholders in the network. Krishna [9] pointed that relational capital refers to the capital rooted in the relationship, emphasizing factors such as attitudes, norms, beliefs, and values. Most scholars believe that relational capital is an important part of social capital, which has a certain influence on the perception and behavior of individuals in the organization [10,11]. Research has verified the role of relational capital in promoting individual performance. When individuals perceive higher trust and identification, they show higher work commitment [12]. The research of Chen et al. [13] found that relational capital could stimulate team creativity, which in turn helped promote team innovation. The research of Ortiz [14] examined the correlation between relational capital and the individual’s knowledge recognition ability and knowledge acquisition ability. Relational capital contributes to knowledge sharing within the organization. Empirical research has verified the positive influence of relational capital on innovation vitality and innovation efficiency [15]. Relational capital can also promote knowledge transfer and value realization within the organization, which in turn has a certain stimulating effect on the overall performance of the organization [16]. Most of the research on relational capital is currently concentrated in the field of knowledge management.

Psychological security is regarded as a feeling of confidence, security and freedom that is separated from fear and anxiety, and it is the feeling of satisfying a person’s various needs, now and in the future [17]. Kahn [18] believes that psychological security is the perception that individuals can fully express their true self without worrying that such behavior will negatively affect their image, status, and career. The higher an employee’s perception of psychological safety, the more obvious it is that they have a positive perception of their environment, the more comfortable their behavior, and the easier it is to stimulate off-role behaviors [19]. Cong and An [20] believe that psychological security is a subjective feeling at the individual level. It is the subjective perception of dangers or risks that may appear around and threaten the body and mind, and it is the perception of whether they have the ability to deal with dangers or risks. This is mainly reflected in the sense of certainty and control. Therefore, psychological security has two main sources: the perception of whether the environment is safe and the judgment of whether one has the ability to cope with change [21]. Enterprises need to pay attention to employees’ psychological safety perception, which is of great significance to the healthy development of the organization [22]. Interpersonal trust and support in an organization can help improve psychological security [18]. High-quality interpersonal relationships can promote learning within the organization through the role of psychological security [23]. If colleagues maintain a relationship of mutual trust and support and communicate frequently at work, employees’ psychological security will also be enhanced [24]. The employee’s perceived relationship capital can promote psychological security. Therefore, we propose Hypothesis 1:

**Hypothesis** **1.**
*Relational capital has a significant positive impact on psychological safety.*


### 2.2. The Mediating Role of Psychological Security

Post-traumatic growth is the positive aspect that an individual exhibits in the face of a traumatic event. It is the positive and positive psychological change experienced by the individual in the process of fighting a traumatic event [25]. The research pointed out that the post-trauma growth of an individual can be found in the improvement of interpersonal relationships, the possibility of obtaining new opportunities, the change of life philosophy, self-growth (confidence, self-efficacy, and acquisition of new coping styles) and spiritual development. Trauma is an unavoidable event in an individual’s life. The individual’s reaction after experiencing a traumatic event is reflected in multiple levels and multiple aspects. It is not only limited to negative health reactions, but also manifested in a series of positive cognitive modes, such as self-awareness and value change [26]. The positive and negative effects of trauma can coexist. Therefore, post-trauma growth is a very normal phenomenon. There have been a large number of studies that have confirmed the existence and influencing factors of post-traumatic growth. From an individual perspective, when their parents are disabled at an early age, individuals become more independent and more responsible for the family [27]. Women are more likely to develop post-traumatic growth than men, and as they grow older, post-traumatic growth is more pronounced [28]. Personality traits are an effective predictor of post-traumatic growth. Individuals with high extroversion, optimistic personality, self-control, and self-efficacy can effectively predict the occurrence of post-traumatic growth [29]. From the view of the trauma source, trauma related to natural disasters is more likely to stimulate the post-traumatic growth of individuals than trauma by human causes [30]. The study of Milam et al. [31] found that when individuals had religious beliefs, they were more likely to have a higher level of post-traumatic growth in the face of trauma. Social support can promote post-traumatic growth. The support of family, friends, teachers and other social forces can make individuals have a higher sense of psychological security, and are more likely to develop post-traumatic growth [32]. Brown [33] pointed out that psychological security can positively predict employees’ work behavior and increase work involvement. Relational capital itself is a kind of social support. Organization members trust each other, adhere to the principle of mutual benefit, help and support each other, and have a sense of belonging to the organization, which will enhance their confidence in coping with difficulties and trauma, and strengthen their individual resistance to pressure. Individuals will have positive growth after trauma, recognize themselves and accept themselves. Therefore, we propose research Hypothesis 2:

**Hypothesis** **2.**
*Relational capital affects post-traumatic growth through psychological security. Psychological security plays a mediating role between relational capital and post-traumatic growth.*


### 2.3. The Moderating Effect of Work Meaning

Work meaning is the value judgment that an individual makes on the purpose of their work, according to his own values and value standards, which reflects the matching level between the individual’s values, beliefs and job role [34]. Employees evaluate organizational goals according to their personal judgment standards. Employees who believe that their work is meaningful and valuable are more willing to work hard and use the autonomy granted by the organization to complete organizational goals [35]. The study of Mok and Au-Yeung [36] showed that teamwork spirit can positively affect employees’ perceived work meaning. Information sharing within the organization helps employees realize the meaning of work, strive to make their own work goals consistent with the organization’s development goals, and put forward valuable suggestions or opinions for the organization. Studies have found that a high-quality leader–member exchange relationship can help enhance employees’ perception of job meaning [37]. The research of Zhang and Bartol [38] pointed out that when employees think that their work is meaningful and have high autonomy, their innovative behaviors are more frequent. The employee’s work meaning perception will increase the employee’s identification and loyalty to the organization [39]. Li Chaoping [40] verified through empirical research that job meaning has a significant positive impact on employees’ organizational commitment and job satisfaction, and negatively impacts job burnout. When employees’ sense of work meaning is higher, it means that employees’ recognition of their work and their sense of satisfaction at work is stronger. If, at the same time, employees can perceive trust from their colleagues and experience reciprocity and a strong sense of belonging at work, meaningful work can further enhance their psychological security. When experiencing high-pressure work challenges, an individual’s perception of the meaning of work and the recognition of work goals can enhance their courage and enthusiasm to cope with difficulties, and it is easier to form self-growth in this process. Therefore, we propose research Hypothesis 3 and research Hypothesis 4:

**Hypothesis** **3.**
*Work meaning has a moderating effect on the relationship between relational capital and psychological security, and high work meaning can enhance the positive effect of relational capital on psychological security.*


**Hypothesis** **4.**
*Work meaning has a moderated mediating effect on the relationship between relational capital, psychological security and post-traumatic growth.*


## 3. Research Design

### 3.1. Procedure

To verify the inner relationship between relational capital, psychological security, post-traumatic growth, and work meaning, as shown in the research model (Figure 1) we collected data in public hospitals of 19 provinces and autonomous regions, including Jiangsu, Sichuan, Ningxia, Guizhou, Yunnan, Beijing, Tianjin and Shanghai from June 2020 to August 2020. A total of 900 questionnaires were distributed and 835 were recovered, with a recovery rate of 92.8%. Of these, 760 questionnaires were valid, and the effective recovery rate was 84.4%.

There were 124 males, accounting for 16.3% of the total sample, and 636 females, accounting for 83.7% of the total sample. The female sample was much higher than the male sample, which is consistent with the gender differences in the medical industry. The respondents were mostly young, with 471 respondents under the age of 30, accounting for 62% of the sample. There were 279 respondents at the age of 30–50, representing 36.7% of the total sample. The majority of the respondents were highly educated. There were 677 people with a bachelor’s degree or above, accounting for 89% of the total sample. There were 182 doctors, representing 23.9% of the total sample, and 578 nurses, representing 76.1% of the sample. A total of 516 were medical staff working in designated hospitals for COVID-19, accounting for 67.9% of the total sample. Another 315 were medical staff directly involved in the treatment of confirmed or suspected patients with COVID-19, accounting for 41.4% of the total sample. During the outbreak of COVID-19, most medical staff carried out intensive work; 419 medical personnel worked more than 10 hours a day, representing 55% of the total sample.

### 3.2. Measurement

This study used mature scales, which was measured by Likert’s five-point scale from 1 completely disagree to 5 completely agree. The questionnaire included demographic variables, relational capital, psychological security, post-traumatic growth, and work meaning. The data were analyzed by SPSS 22, Amos24 and Process V3.4 statistical software, mainly related to descriptive statistics, confirmatory factor analysis, correlation analysis, hierarchical regression analysis, and bootstrap analysis to validate research hypotheses. Relational capital focuses on the particular relations people have, such as respect and friendship, that influence their behaviors. It is through these ongoing personal relationships that people fulfill such social motives as sociability, approval and prestige [7]. Relational capital has three dimensions: trust, reciprocity and identification. Trust refers to an individual’s expectation that members in a virtual community will follow a generally accepted set of values, norms, and principles. Reciprocity refers to the actions that are contingent on rewarding reactions from others and that cease when these expected reactions are not forthcoming. Identification is the process whereby individuals see themselves as one with another person or a group of people (Chiu et al., 2006). The measurement of relational capital is based on the research of Chiu et al. [41], which contains nine items in three dimensions; the internal consistency coefficients of trust, reciprocity and identification are 0.894, 0.801 and 0.821, respectively. Psychological security focuses on a feeling of confidence, security and freedom that is separated from fear and anxiety, and it is the feeling of satisfying a person’s various needs, now and in the future (Authur S. Reber, 1996) [17]. The measurement of psychological security is based on the study of Cong Zhong and An Lijuan [20]; the scale contains fifteen items and the internal consistency coefficient is 0.834. Post-traumatic growth focuses on the positive and positive psychological change experienced by the individual in the process of fighting a traumatic event [25]. The measurement of post-traumatic growth is based on the research of Tedeschi et al. [25], which has 20 items and an internal consistency coefficient of 0.943. Work meaning focuses on the value judgment that an individual makes on the purpose of their work, according to his/her own values and value standards, which reflects the matching level between the individual’s values, beliefs and the job role [34]. Measurement of work meaning is based on the research of Spreitzer [34]. The scale contains three items with an internal consistency coefficient of 0.936.

## 4. Results

### 4.1. Confirmatory Factor Analysis

First of all, we use confirmatory factor analysis to test the validity of the data obtained from this survey and verify whether there is a serious homologous bias problem. Based on the six-factor model (trust, reciprocity, identification, psychological security, post-traumatic growth, and work meaning), we compare the model fit of six factors with that of five-factor, four-factor, three-factor, and one-factor models. As shown in Table 1, the overall model fit of the six-factor model is the best ($\chi^{2}$/df = 1.846, TLI = 0.987, CFI = 0.992, RMSEA = 0.018). The model fit of the single-factor model ($\chi^{2}$/df = 20.513, TLI = 0.722, CFI = 0.790, RMSEA = 0.076) is not within the acceptable range, so the homologous deviation of this study is not serious and further statistical analysis can be carried out.

### 4.2. Correlation Analysis

Through descriptive statistical analysis, we can understand the perceived level of relational capital, psychological security, post-traumatic growth, and work meaning in this survey with the results shown in Table 2. The medical staff surveyed have a moderate level of perceived trust (3.859), reciprocity (4.084), and identification (3.852). Reciprocal perception is the most evident in the work nature of doctors and nurses because the treatment of any patient requires a team effort, which is a process of communicating and learning from each other.

Respondents have a stronger sense of psychological security (4.022) and a sense of meaning at work (3.926). Doctors and nurses have a high reputation in society, especially in the face of an outbreak, and their work is highly valued, which helps to stimulate professional well-being and satisfaction. In a high-pressure work environment, doctors and nurses have a moderate degree of post-traumatic growth (3.980 ); many respondents in this survey said that because of COVID-19, they have rethought the value of life. Despite the unprecedented challenges, they have experienced a degree of positive change in their relationships and personal development that has made them more value what they have now.

The results of the correlation analysis show that trust, reciprocity, identification, psychological security, post-traumatic growth and work meaning are significantly related with each other after gender, age, education, position, and seniority have been controlled. Trust (0.518 **), reciprocity (0.565 **) and identification (0.493 **) all have a significant positive effect on psychological security, indicating that when individuals can trust each other and help each other, and have a higher sense of identification, psychological security is also stronger; therefore, relational capital can promote the formation of individual psychological security. Trust (0.539 **), reciprocity (0.555 **), and identification (0.494 **) all have significant positive effects on post-traumatic growth. There is a significant positive correlation between psychological security and post-traumatic growth (0.719 **), which indicates that the formation of psychological security is essential for the emergence of individual post-traumatic growth, and that the stronger the psychological security, the more likely it is for the individual to experience post-traumatic growth.

### 4.3. Mediating Effect

The mediating effect of psychological security in the relations between relational capital and post-traumatic growth is verified by the hierarchical regression. Taking gender, age, education, position, seniority as the control variables, we discuss the mediating effect of trust, reciprocity and identification respectively in relational capital. The hierarchical regression results are shown in Table 3: Models 1–3 examine the mediating effect of psychological security between trust and post-traumatic growth. After confirming the significant influence of trust on post-traumatic growth (0.440 ***) and psychological security (0.442 ***), when psychological security as a mediating variable enters the regression equation, the effect of the mediating variable is significant (0.601 ***). Although the relationship between trust and post-traumatic growth is still significant, the effect is weaker (0.176 ***); therefore, psychological security has a partial mediating effect between trust and post-traumatic growth. Models 4–6 examine the mediating effect of psychological security between reciprocity and post-traumatic growth. After confirming the significant influence of reciprocity on post-traumatic growth (0.521 ***) and psychological security (0.552 ***), when psychological security as a mediating variable enters the regression equation, the effect of the mediating variable is significant (0.596 ***). Although the relationship between reciprocity and post-traumatic growth is still significant, the effect is weaker (0.192 ***). Therefore, psychological security has a partial mediating effect between reciprocity and post-traumatic growth. Models 7–9 examine the mediating effect of psychological security between identification and post-traumatic growth. After confirming the significant influence of identifications to post-traumatic growth (0.438 ***) and psychological security (0.447 ***), when the psychological security as a mediating variable enters the regression equation, the effect of the mediating variable is significant (0.624 ***). Although the relationship between identification and post-traumatic growth is still significant, the effect is weaker (−0.160 ***). Therefore, psychological security has a partial mediating effect between identification and post-traumatic growth. Through hierarchical regression, Hypothesis 1 and Hypothesis 2 are validated.

### 4.4. Moderating Effect

The process procedure is used to verify the moderating effect of work meaning between relational capital and psychological security, and the moderated mediating effect of work meaning between relational capital, psychological security and post-traumatic growth. The results are shown in Table 4.

First of all, we discuss the moderating effect of work meaning between trust and psychological security. At a high standard of work meaning, the 95% confidence interval of the moderating effect between trust and psychological security (b = 0.2386, SE = 0.0394) is [0.1612, 0.3159], and the interval does not pass zero, which means that work meaning at a high level has a moderating effect between trust and psychological security, namely, a high level of work meaning can enhance the role of trust in the promotion of psychological security. At a low level of work meaning, the 95% confidence interval of the moderating effect between trust and psychological security (b = 0.1052, SE = 0.0324) is [0.0416, 0.1678], and the interval does not pass zero, which means that work meaning at a low level has a moderating effect between trust and psychological security, namely, low level of work meaning inhibits the role of trust in promoting psychological security. When work meaning is taken into account in the mediating effects of trust between psychological security and post-traumatic growth (b = 0.0456, SE = 0.0192), the interval under the 95% confidence level is [0.0084, 0.0839] and the interval does not pass zero, so work meaning plays a moderated mediating effect on the relationship between trust, psychological security and post-traumatic growth.

Secondly, we discuss the moderating effect of work meaning between reciprocity and psychological security. At a high standard of work meaning, the 95% confidence interval of the moderating effect between reciprocity and psychological security (b = 0.1919, SE = 0.0425) is [0.1086, 0.2763], and the interval does not pass zero, which means that work meaning at a high level has a moderating effect between reciprocity and psychological security, namely, a high level of work meaning can enhance the role of reciprocity in the promotion of psychological security. At a low level of work meaning, the 95% confidence interval of the moderating effect between reciprocity and psychological security (b = 0.0706, SE = 0.0375) is [−0.0030, 0.1442], and the interval does pass zero, which means that work meaning at a low level has not a moderating effect between reciprocity and psychological security, namely, the effect of low working meaning on the intrinsic relationship between reciprocity and psychological security does not exist. When work meaning is taken into account in the mediating effects of reciprocity between psychological security and post-traumatic growth (b = 0.0434, SE = 0.0218), the interval under the 95% confidence level is [0.0005, 0.0867] and the interval does not pass zero, so work meaning plays a moderated mediating effect on the relationship between reciprocity, psychological security and post-traumatic growth.

Finally, we discuss the moderating effect of work meaning between identification and psychological security. At a high standard of work meaning, the 95% confidence interval of the moderating effect between identification and psychological security (b = 0.0762, SE = 0.0260) is [0.0253, 0.1272], and the interval does not pass zero, which means that work meaning at a high level has a moderating effect between identification and psychological security, namely, a high level of work meaning can enhance the role of identification in the promotion of psychological security. At a low level of work meaning, the 95% confidence interval of the moderating effect between identification and psychological security (b = 0.0521, SE = 0.0317) is [−0.0101, 0.1143], and the interval does pass zero, which means that work meaning at a low level does not have a moderating effect between identification and psychological security, namely, the effect of low working meaning on the intrinsic relationship between identification and psychological security does not exist. When work meaning is taken into account in the mediating effects of identification between psychological security and post-traumatic growth (b = 0.0309, SE = 0.0228), the interval under the 95% confidence level is [−0.0162, 0.0738] and the interval does pass zero, so work meaning does not play a moderated mediating effect on the relationship between identification, psychological security and post-traumatic growth. Hypothesis 3 and Hypothesis 4 are partially validated.

## 5. Conclusions and Discussions

The war against the epidemic has not been declared a final victory, and countless medical staff and society are continuing to fight with COVID-19. Extensive research has shown that traumatic experiences might influence physical and mental state, even resulting in post-traumatic stress disorder. At the same time, positive psychology points out that many individuals have the opportunity to grow in the process of crisis response and treatment. Our study explains a possible cause of individual post-traumatic growth from the perspective of relational capital.

Based on statistical analysis, it is proved that relational capital can promote the realization of individual post-traumatic growth through psychological security, that is, when the individual perceives mutual trust within the organization, stays in a reciprocal environment, and has a strong sense of belonging, they have higher psychological security; this also can help the individual to rebuild values, enhance the individual’s response ability to the crisis and find new development opportunities, which will lead to individual growth. Work meaning plays a regulating role between relational capital and psychological security. When the individual perceives a higher level of work meaning, the promoting effect of trust, reciprocity and identification on psychological security is enhanced. However, when at the low level of work meaning, only the effect of trust on the promotion of psychological security is enhanced, while the influence of reciprocity and identification on psychological security are not affected.

The results of the study may reflect some characteristics of doctors and nurses. Compared with other social occupational groups, their overall work value perception is high, especially in an outbreak environment. From the analysis of their work characteristics, we can see that most medical staff are working in teams. Reciprocity is a basic requirement at work, and their sense of belonging is relatively high. At the same time, due to the continuous progress of medical technology, the competency requirements of doctors and nurses are increasing and the internal competition is also fiercer, which may lead to a challenge in the trust between each other. This study only confirms the existence of the moderated mediating effects of work meaning on the relationship between trust reciprocity, psychological security and post-traumatic growth.

Whether it is natural or man-made, adversity and trauma are inevitable for human beings. How to recover from trauma and make life more active is the most important issue in trauma research. Relational capital as a social support is essential for psychological rehabilitation; in particular, in Chinese culture, people pay more attention to maintaining good relations. Therefore, when encountering major setbacks or crises, relational capital is especially important so that individuals can respond and recover more quickly after the crisis. Organizations should strive to promote the formation of an internal trust mechanism and atmosphere through cultural construction and value management; they should also help members form mutually beneficial working relations through interactive learning, management by objective, knowledge sharing, etc., and enhance their sense of belonging through organizational compensation, special ceremonies, etc. Relational capital gives employees more confidence to cope with high pressure and uncertainty, and can boost their future growth. In the recruitment process, selecting employees who are more recognized for their work, and strengthening the individual’s sense of work value through various publicity and training activities will also help the occurrence of post-traumatic growth. It is not an accidental phenomenon that individuals have positive feedback and growth after experiencing major crises and traumas. This study only explores the role of relational capital, but there are many individual traits, group interaction factors, organizational supportive factors that may affect the occurrence of post-traumatic growth, which also need to be further studied in future research.

## Figures and Tables

**Figure 1 ijerph-18-07362-f001:**
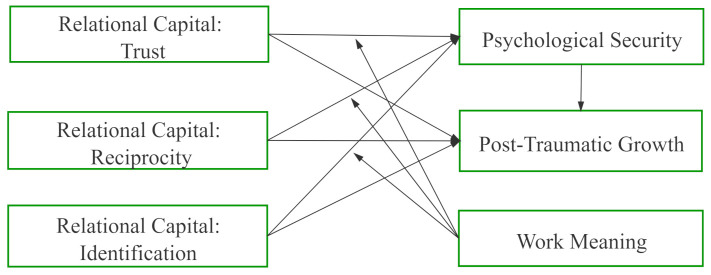
Research Model.

**Table 1 ijerph-18-07362-t001:** Confirmatory factor analysis.

Model	$\chi^{2}$	*df*	$\chi^{2}$ */df*	TLI	CFI	RMSEA
Six-Factor Model	230.722	125	1.846	0.987	0.992	0.018
Five-Factor Model	347.659	131	2.654	0.975	0.983	0.020
Four-Factor Model	450.671	135	3.338	0.965	0.975	0.036
Three-Factor Model	731.929	136	5.382	0.934	0.953	0.057
Single-Factor Model	2646.192	129	20.513	0.722	0.790	0.076

**Table 2 ijerph-18-07362-t002:** Means, standard deviations and correlations (N = 760).

	Mean	SD	1	2	3	4	5	6	7	8	9	10
Gender	1.84	0.370
Age	1.51	0.740	−0.262 **
Education	2.07	0.575	−0.297 **	0.360 **
Position	1.76	0.427	0.512 **	−0.465 **	−0.585 **
Tenure	1.95	1.009	−0.183 **	0.831 **	0.290 **	−0.324 **
TRU	3.86	0.858	0.010	−0.170 **	−0.100 **	0.168 **	−0.146 **
REC	4.08	0.747	0.013	−0.159 **	−0.091 *	0.157 **	−0.130 **	0.713 **
IDE	3.85	0.789	−0.039	−0.047	−0.099 **	0.142 **	−0.045	0.616 **	0.533 **
PS	4.02	0.726	−0.070	−0.031	−0.030	0.068	−0.028	0.518 **	0.565 **	0.493 **
PTG	3.98	0.723	−0.023	−0.127 **	−0.085 *	0.147 **	−0.103 **	0.539 **	0.555 **	0.494 **	0.719 **
WM	3.93	0.813	−0.035	−0.178 **	−0.135 **	0.164 **	−0.181 **	0.590 **	0.531 **	0.607 **	0.674 **	0.641 **

Note: N = 760; ** *p* < 0.01, * *p* < 0.05; TRU, trust; REC, reciprocity; IDE, identification; PS, psychological security; PTG, post-traumatic growth; WM, work meaning.

**Table 3 ijerph-18-07362-t003:** Mediating effect.

	Model 1	Model 2	Model 3	Model 4	Model 5	Mdoel 6	Model 7	Model 8	Model 9
	PTG	PS	PTG	PTG	PS	PTG	PTG	PS	PTG
Gender	−0.156 *	−0.171 *	−0.053	−0.165	−0.174 *	−0.061	−0.123	−0.136	−0.039
Age	−0.020	0.057	−0.054	−0.012	0.068	−0.052	−0.094	−0.018	−0.083
Education	0.006	0.010	0.000	0.004	0.008	−0.001	0.031	0.035	0.009
Position	0.160 *	0.097	0.102	0.168	0.096	0.111	0.140	0.072	0.095
Tenure	0.003	0.013	0.003	−0.005	−0.008	0.000	0.005	0.002	0.004
TRU	0.440 ***	0.442 ***	0.176 ***						
REC				0.521 ***	0.552 ***	0.192 ***			
IDE							0.438 ***	0.447 ***	0.160 ***
PS			0.601 ***			0.596 ***			0.624 ***
$\Delta R^{2}$	0.229 ***	0.277 ***	0.562 ***	0.317 ***	0.328 ***	0.558 **	0.259 ***	0.247 ***	0.554 ***
F	53.487	48.030	138.079	58.312	61.331	135.575	43.294	41.113	133.626

Note: N = 760; *** *p* < 0.001, ** *p* < 0.01, * *p* < 0.05; TRU, trust; REC, reciprocity; IDE, identification; PS, psychological security; PTG, post-traumatic growth; WM, work meaning.

**Table 4 ijerph-18-07362-t004:** Moderating effect and moderated mediating effect.

	Moderating Effect				Moderated Mediating Effect			
**Variable**	**Effect**	**SE**	**95% Confidence Interval**		**Effect**	**SE**	**95% Confidence Interval**	
			**Upper**	**Lower**			**Upper**	**Lower**
TEU	0.1052	0.0324	0.0416	0.1678	0.0456	0.0192	0.0084	0.0839
	0.2386	0.0394	0.1612	0.3159				
REC	0.0706	0.0375	−0.0030	0.1442	0.0434	0.0218	0.0005	0.0867
	0.1919	0.0425	0.1086	0.2763				
IDE	0.0521	0.0317	−0.0101	0.1143	0.0309	0.0228	−0.0162	0.0738
	0.0762	0.0260	0.0253	0.1272				

Note: TRU, trust; REC, reciprocity; IDE, identification; PS, psychological security; PTG, post-traumatic growth; WM, work meaning.

## Data Availability

The data presented in this study are available on request from the corresponding author.

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
