# Peer review of "Relational Capital and Post-Traumatic Growth: The Role of Work Meaning"

_ijerph, 2021, doi:10.3390/ijerph18147362_

Round 1
Reviewer 1 Report
The manuscript was intermittently difficult to follow. Specific suggestions include:
Changing-
Page 2 – “attitude s” to “attitudes”
Page 3 – “The work meaning” to “Work meaning”
Page 9 – “medical staff and all the society are continuing to fight with covid-19” to “medical staff and society are continuing to fight the effects of covid-19”
Page 9 – “Extensive research that has shown that traumatic experiences might influence physical and mental state, even resulting in post-traumatic stress disorder.” to “Extensive research has shown that traumatic experiences might influence physical and mental state, even resulting in post-traumatic stress disorder.”
Page 9 - But at the same time positive psychology also points out that many individuals also get the opportunity to grow in the process of crisis response and treatment…” to “But at the same time positive psychology points out that many individuals get the opportunity to grow in the process of crisis response and treatment…”
Page 9 – “…this study explains a possible cause of individual post-traumatic growth from the perspective of relational capital.” To “Our study explains a possible cause of individual post-traumatic growth from the perspective of relational capital.”
Page 10 – “China culture” to “Chinese culture”
Page 12 – “Post-traumatic growth is the positive aspect that an individual exhibit in the face of a traumatic event.” To “Post-traumatic growth is the positive aspect that an individual exhibits in the face of a traumatic event.”
Page 12 – “natural caused trauma” – consider “trauma related to natural disasters” or "trauma caused by natural disasters".
It would also be useful to define the types of relationship capital, knowledge recognition ability, and knowledge acquisition ability.
Author Response
Dear Reviewer,
We are deeply grateful for the suggestions and ideas contributed by the reviewer’s intelligent work. Very constructive comments were made to our author team. We were discussing and rethinking the paper from different perspectives in the past week. We have learnt not only academic insight but also literature skills to establish a comprehensive academic research conclusion. In response to the reviewer’s comments, after careful consideration by the authors, the relevant parts of this article have been revised and some materials have been added.
Please see the attachment.

Reviewer 2 Report
The topic of the manuscript is actual, the treatment of post-covid issues will have an increasingly important role in any scientific area. This paper is a good contribution to the knowledge base in the field. The data collection and processing is acceptable; it is in line with recent papers. The goals are clear, description of the content is correct. After a plagiarism check, I can state the this is an original work.
Still, I have to ask for a major revision of the paper. There is one reason for this. The literature background includes only pre-covid sources. Checking the scientific databases, I found papers in the field of the manuscript. A review of these is necessary, and either confirmation of their conclusions or finding the gaps are required to justify the own results.
Author Response

(The authors gave the same response as above.)
